# LWIR Lateral Effect Position Sensitive HgCdTe Photodetector at 205 K

**DOI:** 10.3390/s23104915

**Published:** 2023-05-19

**Authors:** Jarosław Pawluczyk, Mateusz Żbik, Józef Piotrowski

**Affiliations:** 1Institute of Applied Physics, Military University of Technology, 00-908 Warsaw, Poland; 2VIGO Photonics S.A., 05-850 Ozarow Mazowiecki, Poland; 3Institute of Electronic Systems, Warsaw University of Technology, 00-665 Warsaw, Poland

**Keywords:** LWIR, HgCdTe, PSD, position-sensing detector, photodetector, lateral effect, tetra-lateral, PIN photodiode, Peltier-cooled

## Abstract

We describe in detail the construction and characterization of a Peltier-cooled long-wavelength infrared (LWIR) position-sensitive detector (PSD) based on the lateral effect. The device was recently reported for the first time to the authors’ knowledge. It is a modified PIN HgCdTe photodiode, forming the tetra-lateral PSD, with a photosensitive area of 1 × 1 mm^2^, operating at 205 K in the 3–11 µm spectral range, capable of achieving a position resolution of 0.3–0.6 µm using 10.5 µm 2.6 mW radiation focused on a spot of the 1/e^2^ diameter 240 µm, with a box-car integration time of 1 µs and correlated double sampling.

## 1. Introduction

A position-sensing detector (PSD) is a photosensor that detects the position of a light spot on its surface. A PSD enables precise and fast beam alignment and tracking in a variety of applications [1], such as active beam stabilization, optical remote-control devices, infrared (IR) range finders, target detection and tracking, missile interception, IR countermeasures, optical switches, and measurements of vibrations, distortions, and lens refraction/reflection.

The motivation for the responsivity spectral range of up to 10–11 μm, embracing a significant portion of the LWIR band (wavelength λ = 8–14 µm), includes detection and tracking of distant objects, where the LWIR may be preferred to the MWIR (λ = 3–5 µm) and a shorter wavelength due to less aerosol scattering and better performance in environments with dust, fog, or winter haze; higher immunity to atmospheric turbulence; and reduced sensitivity to solar glints and fire flares [2] (p. 5).

LWIR PSDs can also be applied in positioning systems with detection in the LWIR band, especially when LWIR laser beams are used, e.g., in optical communication in free space, in industry using high-power CO_2_ lasers for hard-material treatment, or in nondestructive inspection and metrology tools. There are numerous medical, industrial, and military applications in the LWIR range [3].

Owing to the frequent requirement for fast and accurate position detection and tracking, PSDs based on fast photon detectors, such as photodiodes, are often preferred to thermal PSDs–quadrant detectors or dense arrays, the acceptable sensitivity of which is achieved at the expense of their slow response times of the order of milliseconds and more [4] (p. 57) [5].

PSDs with the best parameters in terms of position sensitivity and linearity use the lateral photoelectric effect (L-PSD devices) [1,6,7,8]. L-PSDs are single-element detectors of various types and materials, p-n photodiodes or heterostructures, also field-effect phototransistors, where the built-in or externally supplied electric field, perpendicular to the substrate, enables the separation of photogenerated electrons and holes, which next spread laterally in a resistive layer, which will be described later on [7,8,9].

The lateral photoelectric effect was originally described by Schottky in 1930 [10], rediscovered and further elaborated by Wallmark in 1957 [11], and described by Lucovsky in 1960 with analytical equations [12], which help interpret the photoelectric effects observed in nonuniformly irradiated p-n junctions. Developments in the field have since been achieved by many researchers, including the optimization of a PSD construction [13,14] and the shape of electrodes in particular [15].

The main advantage of the L-PSD over photon quadrant detectors (QDs) [16,17], which have been used so far as fast Peltier-cooled PSDs for the LWIR band, is that L-PSD performance is much less dependent on the light-spot intensity distribution. In the case of the L-PSD, the lateral extent of the position measurement span corresponds to the side length of the photosensitive area, while in the case of the QD it corresponds to the diameter of the light spot. Hence, the position linearity is much greater for the L-PSD than for the QD for a given photosensitive area. The L-PSD may also replace a linear array detector, e.g., in compact spectrometers [7].

Previous LWIR L-PSD designs have used cryogenic cooling [6], which is common for LWIR detectors, as it reduces their electric charge thermal generation-recombination (G-R) noise below the level determined by the 300 K thermal background, allowing background-limited performance (BLIP). A cryo-cooler, though, makes the detector assembly bulky, expensive, and often less convenient to operate. Therefore, we decided to explore the possibility of position detection of 10–11 µm radiation at a higher operating temperature, ca. 200 K, achievable by Peltier mini-coolers–small, rugged devices with an operating life of about 20 years and a supply power below 2 W. This is perhaps the first demonstration of such an operation, carried out here on a HgCdTe photodiode with a photosensitive area of 1 × 1 mm^2^, large for this spectral range and operating temperature, at reverse bias 0.16 V/50 mA. However, power-consuming cooling and bias are also not desirable for portable and integrated devices.

The current L-PSD performance for the IR range of λ < 2 µm (SWIR) may be used as a first benchmark for LWIR PSD development. This includes position resolution of about 0.1 µm normalized to a photosensitive area 1 mm in size, with response times on the order of one to several tens of microseconds and excellent position linearity, with nonlinearity < 2%, obtained without cooling at 0 V bias [1,8,18,19].

SWIR detectors, including L-PSDs, benefit from wide-bandgap semiconductors, used as radiation absorbers. They require neither cooling nor bias to achieve BLIP. SWIR L-PSDs have been commercially available for several decades at a moderate price [9,19]. On the other hand, uncooled or Peltier-cooled LWIR photon detectors suffer from huge thermal generation noise. Their sensitivity is several orders of magnitude lower than that of the wide-gap devices. This is true even for BLIP LWIR detectors, because of the low energy of LWIR photons, about 0.1 eV, comparable to the constant kT = 0.026 eV of the Boltzmann distribution at T = 300 K, resulting in high (300 K) thermal background flux and noise in this spectral range. This is exacerbated by the contribution of nonradiative thermal G-R mechanisms, still playing a role in many LWIR uncooled or Peltier-cooled detectors. The nonradiative high thermal G-R rate in LWIR absorbers results from the low energy of transition of electrons between energy levels, corresponding to the low energy of the absorbed LWIR photons. The thermally generated dark currents of uncooled or Peltier-cooled LWIR photon detectors can be very high if they are not blocked by special barriers in bandgap-engineered structures.

Figure 1 schematically illustrates the principle of operation of the lateral effect photodiode used here. It is a modified PIN photodiode made of LWIR HgCdTe. The inter-band absorption of IR in the absorber layer generates electron–hole pairs, collected as a photocurrent. The device is shaped as a mesa structure, where the lower cathode layer forms the resistive layer acting as the current divider, while the upper anode layer is a common contact. The photocurrent is divided between four cathode metal electrodes in proportion to the resistances of the photocurrent paths between the illuminated region and the electrodes in one resistive layer. Thus, this is the tetra-lateral type (TL-PSD) of L-PSD [13]. In the applied configuration, a pair of cathode electrodes is extended along the opposite edges of the mesa structure, parallel to each other and perpendicular to the same electrodes of the second pair.

With Peltier modest cooling to 205 K, the photodiode resistance at 0 V bias is low (<1 Ω) and dominated by the parasitic series resistance, due to the high thermal G-R. The device must be reverse biased to maximize the differential impedance of the photodiode, which is critical to the performance–sensitivity and linearity–of the LWIR L-PSD.

Assuming perfect position linearity, the transverse position (X, Y) of the light spot gravity center is determined by subtracting the signals from the opposite cathode electrodes (channels) and dividing this difference by the sum of these signals (adapted with permission from Ref. [20]; copyright: 2022, SPIE):(1)$$X_{PSD}=\frac{i_{B}-i_{C}}{i_{B}+i_{C}}\times k_{X}=\frac{U_{B}-U_{C}}{U_{B}+U_{C}}\times k_{X}$$
(2)$$Y_{PSD}=\frac{i_{A}-i_{D}}{i_{A}+i_{D}}\times k_{Y}=\frac{U_{A}-U_{D}}{U_{A}+U_{D}}\times k_{Y}$$
where X_PSD_ and Y_PSD_ are the Cartesian coordinates of the radiation beam spot position determined from the TL-PSD signals, i.e., the U_A_, U_B_, U_C_, and U_D_ voltages at respective channel preamplifier outputs. The preamplifiers integrate the photocurrent, giving the output voltage proportional to the input radiation pulse energy. The scale factors k_X_ and k_Y_ are proportional to the resistance length L ≈ L_X_ ≈ L_Y_ ≈ 1 mm (Figure 1). If the dark current of the photodiode is significant, k_X_ and k_Y_ are also affected by the dark current and energy of the radiation pulses. In this case, for k_X_ and k_Y_ to be constant, the radiation pulse energy should be constant too [7]. The resistance lengths L_X_ and L_Y_ represent the linear size of the detector photosensitive area in the X and Y directions, respectively, where the TL-PSD signals change with the respective X and Y position shifts, preferably in linear proportion.

The nonuniformities in the photodiode material layers influence the distribution of the photocurrent between the cathode electrodes and thus cause an error in the light spot position measurement. The position signal error can be corrected using calibration and postprocessing, providing the position measurement noise is low enough. However, time-consuming corrections are not desired in view of the required fast position signal response. Thus, the manufacturing process quality, ensuring the uniformity of the material layers and interfaces, is a key factor.

Even assuming the uniform material layers and the ohmic contacts of the electrodes to the resistive layer, the position linearity and the resulting error of the position estimation using Formulas (1) and (2) worsen with an increase in the size of the light spot and the distance of its gravity center from the middle of the detector active area, depending on the shape and configuration of the electrodes on the resistive layer and the contribution of the series impedance to the total impedance of the detector. Nonlinear effects and corrections realizable by simple fast analog electronics are described in [21].

The position resolution of a PSD is the minimum detectable displacement of a light spot [7,9]. This means that it is related to the signal-to-noise ratio of the detector and read-out electronics. The position resolution can be assessed using the standard deviations $\bar{x_{nPSD}}$ and $\bar{y_{nPSD}}$ of the position coordinates X_PSD_ and Y_PSD_, respectively (adapted with permission from Ref. [20]; copyright: 2022, SPIE):(3)$$\bar{x_{nPSD}}=2k_{X}\frac{\sqrt{U_{B}^{2}{\bar{u_{nC}}}^{2}+U_{C}^{2}{\bar{u_{nB}}}^{2}}}{{\left(U_{B}+U_{C}\right)}^{2}}$$
(4)$$\bar{y_{nPSD}}=2k_{Y}\frac{\sqrt{U_{A}^{2}{\bar{u_{nD}}}^{2}+U_{D}^{2}{\bar{u_{nA}}}^{2}}}{{\left(U_{A}+U_{D}\right)}^{2}}$$
where $\bar{u_{nA}}$, $\bar{u_{nB}}$, $\bar{u_{nC}}$, and $\bar{u_{nD}}$ are the root-mean-square (rms) noises at the respective channel outputs. It is assumed that the noises from the individual channels are uncorrelated with each other.

## 2. Chip Construction and Assembly

A (100)-oriented Hg_1−x_Cd_x_Te heterostructure, containing a lightly doped p^−^-type narrow-gap absorber with the material composition x = 0.2, optimized for position detection of the 10–11 µm radiation at 205 K and sandwiched between wide-gap heavily doped contact layers [22], was grown by metalorganic chemical vapor deposition (MOCVD) on a CdTe-buffered semi-insulating (100) GaAs substrate. The structure was doped with iodine and arsenic as well-behaved and stable donor and acceptor dopants, respectively, by in situ incorporation during growth. The absorber thickness was reduced to about 2 µm in order to increase the differential impedance of this large LWIR photodiode. The input radiation beam essentially passed twice through the absorber, due to the internal reflection from the top of the detector mesa structure, covered with the indium electrode.

Chips of the TL-PSD devices were fabricated, flip-chip-bonded to sapphire carriers, mounted on a Peltier cooler, wired with thin gold leads to output pins, sealed in hermetic packages, integrated with read-out electronics, and scanned with a radiation spot. The chip-level layout used is shown in Figure 2, Figure 3 and Figure 4. Figure 2b provides a top-view micrograph of the fabricated chip. The direction of illumination is indicated in Figure 3. Table 1 schematically presents a diagram of the layers in the mesa structure, with their donor N_D_ and acceptor N_A_ doping concentrations, compositions, and thicknesses, in more detail than in the drawings. In the first column, n^+^ and p^+^ denote donor and acceptor, respectively, heavy doping in a narrow-gap HgCdTe; N^+^ and P^+^ denote donor and acceptor, respectively, heavy doping in a wide-gap HgCdTe; p^−^ denotes acceptor light doping of the narrow-gap absorber. The net doping N_A_ = 2 × 10^15^ cm^−3^ of the absorber is a result of subtraction of the background donor doping—ca. 8 × 10^14^ cm^−3^–from the actual arsenic concentration. The wide-gap layers, N^+^–contact for electrons–and P^+^–contact for holes—are interfaced to the absorber through the graded-gap HgCdTe layers. The CdTe passivation at the edge of the mesa is deposited by sputtering.

The photographs in Figure 5 present the hermetically sealed detector package (Figure 5a); its inner parts (Figure 5b) with the TL-PSD dice, which is flip-chip indium bump bonded to the chip carrier, on the cooler cold finger, where the cooler is soldered to a TO8 header; and the detector shortly connected to the electronics package (Figure 5c).

## 3. Dark Current Sources

Unfortunately, the radiative generation from the 300 K background is negligible compared to the non-radiative G-R components in this device. The dark current is suppressed with cooling, and the reverse bias further reduces the carrier concentrations below their thermal equilibrium levels thanks to a phenomenon specific to infrared narrow-gap semiconductor detectors—suppression of the Auger generation by extraction and exclusion of electric charge carriers with the reverse bias [23].

The dark current sources relevant to this detector are as follows [4] (pp. 247–260), [24] (pp. 82–86):Diffusion currents from the quasi-neutral (neutral) regions, mainly from the depleted in carriers but still neutral absorber volume, with their thermal G-R components:○Auger 7 and Auger 1; in HgCdTe, the former dominates over the latter if the concentration of holes is more than about 10 times greater than that of electrons [24] (pp. 34, 136, 137), [25] (pp. 83–85);○Shockley–Read–Hall (SRH);Currents from the space-charge (depletion) regions around (metallurgical) junctions:
○SRH (thermal G-R current in the space-charge region [26]);○Trap-assisted tunneling (TAT) [27,28], with nearly linear dependence on reverse bias voltage < 1 V;○Band-to-band tunneling (BTB), with voltage-dependence much stronger than linear;Possible components associated with injection to the absorber from the contact regions, mainly over the P^+^ barrier;Surface generation: SRH, tunneling, and conduction channels:○at the edge of the mesa;○along dislocations intercepting the junctions [29,30], i.e., reaching the space-charge regions in the absorber interfaces;Photoelectric gain mechanisms, affecting the dark current, noise, and responsivity but not the detectivity (sensitivity) of the detector [31,32];Internal radiative G-R is not important due to the photon recycling [33].

## 4. Noise Sources

The detector current fluctuates and so generates noise. Four types of noise are of importance in semiconductors [34]:Thermal (Johnson) noise and shot noise, with a constant (white) spectral density within the noise equivalent bandwidth;Random telegraph noise—a noise type that presents as a binary fluctuation in dark current, with a Lorentzian-shaped noise spectral density;1/f noise, with the inverse dependence of the noise spectral power on frequency:
(5)$$\bar{i_{n1/f}^{2}\left(f\right)}=\bar{i_{n1/f}^{2}\left(1\;Hz\right)}/f^{\gamma }$$
where $\bar{i_{n1/f}}$ is the 1/f noise component of the root-mean-square (rms) noise current spectral density, f is the frequency, and γ = 1.0 ± 0.1 in a wide frequency range, after [34]. This is the dominant type of noise in our PSD.

Scientists agree that 1/f noise in semiconductor devices is caused by fluctuations in electrical conductivity but differ on whether carrier concentration fluctuations [35] (pp. 207–228) or carrier mobility fluctuations [34], or both [36,37,38,39], are the cause.

The first paper on 1/f noise in HgCdTe photodiodes was published by Tobin et al. [40] in 1980. The authors found the 1/f noise to be proportional to the surface generation and surface leakage current of implanted n^+^-p MWIR HgCdTe photodiodes for a wide range of operating temperatures, from cryogenic with domination of the surface generation and surface leakage to higher temperatures, where the diffusion dark currents prevailed. The coefficient of proportionality, α, is often referred to as the Tobin coefficient:(6)$$\alpha =\frac{\bar{i_{n1/f}\left(f,T_{d},V_{b}\right)}\times \sqrt{f}}{i\left(T_{d},V_{b}\right)}=\frac{\bar{i_{n1/f}\left(1\;Hz,T_{d},V_{b}\right)}}{i\left(T_{d},V_{b}\right)}$$
where i is the detector overall current or one of its components, T_d_ is the detector operating temperature, and V_b_ is the detector bias voltage. According to Tobin et al., the surface generation and surface leakage could be modulated by fluctuations in the surface potential, thus generating the 1/f noise. The authors were inspired by numerous works on 1/f noise studied in silicon and germanium filaments, p-n junctions [41], and MOS transistors, including McWhorter’s suggestion in 1953 that variations in the occupancy of the slow surface states caused conductivity fluctuations of semiconductor filaments and had a wide uniform spectrum of relaxation times corresponding to the 1/f noise [35] (pp. 207–228). Single-electron slow trapping de-trapping generates random telegraph noise [42]. In photodiodes, the slow trapping events mainly occur in passivation near the semiconductor surface or in the dislocations intercepting the junction, both in depletion regions.

Exemplary 1/f noise calculations for HgCdTe photodiodes were performed, based on McWhorter’s theory [29,43,44,45] or taking into account carrier mobility/scattering fluctuations [38,46]. The dislocations intercepting the junction were found to induce TAT and 1/f noise and are supposed to cause bandgap narrowing and getter impurities that can act as active trap centers [30,32,38,42,46,47,48].

In 2019, the new, more demanding standard was introduced by Teledyne—the MWIR and LWIR HgCdTe reverse biased photodiodes with fully depleted absorbers and close to the 300 K background limited performance [49]. In the fully depleted absorber, dislocations can be inactivated (“frozen”) because of lack of carriers throughout the absorber, even at the highest reported [49] operating temperature of 250 K, as they have been depleted and inactivated so far using cryogenic cooling. Then, the 1/f noise can be greatly reduced, providing the heteropassivation is applied by interdiffusion between the HgCdTe and the CdTe passivation layer during the high-temperature annealing. High-temperature annealing was also found to inactivate defects and dislocations in the active volume of the detector [47].

Another way to avoid or significantly reduce the 1/f noise is to minimize the depletion area using photodiodes optimized for 0 V bias or biased barrier devices, such as nBn [50,51].

## 5. Parameters of the Detector’s Equivalent Circuit

The equivalent circuits treat the TL-PSD like a four-way output current divider with four resistances. Their values can be approximated using the formulas written in Figure 6. The least error of such approximation is obtained at the center position of the light spot, with the radiation distribution approximated by Dirac’s delta function. Figure 7 shows a piece of the equivalent circuit suited for the finite element analysis—two adjacent cells of an array consisting of n × n equal cells. R_p_ denotes the differential parallel resistance and C_p_ denotes the differential parallel capacitance of the photodiode at a given reverse bias. The R_sh_ series resistance is the square sheet resistance of the resistive layer, and R_s com_ is the common series resistance, which is approximately the series resistance of the photodiode excluding the resistive layer. Both models are limited to the small signal case for which the input radiation power and the photocurrent are within the detector linear operation range.

The linearity of the PSD signal versus the input radiation power as well as versus the light spot position and the resolution of the position measurement improve when the differential parallel impedance R_p_‖C_p_ becomes much greater than the series impedance. The resistance of the photodiode was determined from DC measurement of its I-V characteristics, plotted in Figure 8, at the detector operating temperature T_d_ = 205 K. It was important that the measurement was performed with all the A, B, C, and D outputs connected together. A significantly different shape of the characteristics, with an increased contribution of the series resistance, would be obtained if, for example, only one channel was connected to the ammeter and the other three were open.

At the reverse bias V_b_ outside the range of 0.1–0.2 V, the detector resistance dropped below <1 Ω and was dominated by the series resistance. With V_b_ increase to 0.12 V, the dark current became saturated due to the suppression of Auger generation, before the loss of T_d_ stabilization by too much Joule heating could occur. The dark current slightly decreased with V_b_ in the range where the differential resistance was negative and then increased due to TAT, which narrowed the range of high differential resistances to a few millivolts of V_b_. Even infinite differential resistance is theoretically possible, but a maximum of just over 100 Ω has been reached. This was due to the narrow V_b_ range with high differential resistances and the preamplifier input offset voltage, ranging from −0.5 to 0.5 mV [52].

The suppression of Auger generation with reverse bias is possible with low doping of the relatively thin HgCdTe absorber layer. However, in spite of its light p^−^-type doping (Table 1) and small thickness, we did not manage to suppress the electron concentration in the absorber to a negligible value. The minimal dark current at V_b_ ≈ 0.16 V occurred at the level corresponding to the Auger 1 rather than the Auger 7 diffusion current and was close to that predicted by “Rule 07” [53], a metric derived in 2007 for detectors limited by diffusion currents generated in a HgCdTe n-type absorber with extrinsic donor doping 10^15^ cm^−3^. Nevertheless, “Rule 07” has become a popular benchmark for state-of-the-art IR photodetectors based on other technologies, such as nBn and type II strained-layer superlattices.

The surface at the edge of this mesa structure insignificantly contributes to the dark current generation, due to the mesa transverse size being > 100 times greater than the carrier diffusion length. This was confirmed in dark current measurements of photodiodes of variable size from the same wafer. Nevertheless, the surface generation may significantly contribute to 1/f noise, as in [54].

The R_sh_ sheet resistance was checked as the resistance R_C-B_ between the opposite cathode electrodes C and B and as the resistance R_D-A_ between the opposite cathode electrodes D and A, when the anode electrode was electrically open. Figure 9 plots the differential resistances R_C-B_ and R_D-A_ and the currents versus the voltage put between the electrodes. The minimum R_C-B_ and R_D-A_ values were around 0 V and amounted to 5.5 Ω and 5.7 Ω, respectively, close to 5.5 Ω of the R_sh_ measured on the wafer before chip-processing. Assume an even distribution of the dark current I_d_ between the cathode electrodes, I_d/ch_ = 12 mA per one channel using 0.16 V of the reverse bias. During a normal operation of the PSD, when the photosignal varies linearly with the input radiation power, the amplitude of the total photocurrent i_ph_ does not exceed 4 mA, i.e., 1 mA per channel. The R_C-B_ R_C-B_ and R_D-A_ values for ±16 mA, corresponding to the sum of ±12 mA of the dark current and ±4 mA of the photocurrent, are given in Table 2.

The resistance characteristics, exhibiting some nonlinearity and asymmetricity, and the observed differences between the R_C-B_, R_D-A_, and R_sh_ values indicate the possible contribution of the resistance of the metallization contact to the resistive layer. This is comparable to the resistance 0.6 Ω of the 25 µm diameter, 1 cm long gold lead wire and its compression bond used to connect the cathode electrode with the output pin. The sum of the metal–semiconductor contact resistance and that of the lead wire makes about 1 Ω of the detector series resistance at the output of each channel. It does not change with the light spot position and negatively affects the position error and the PSD linearity. Unfortunately, it was small compared to the 8 Ω for the input impedance of the integrating preamplifier, calculated later using Equation (9).

The inductance of the single lead wire can be assumed to be L = 13.3 nH, as the inductance of a straight gold wire 25 µm in diameter and 1 cm long [55,56]. For the applied integration times, which are time intervals τ_CDS_ of the correlated double sampling (CDS; Section 6), τ_CDS1_ = 1 µs and τ_CDS2_ = 0.5 µs, the 3 dB frequencies f of an integrating preamplifier transfer function are those of sinc(π τ_CDS_ f)—the Fourier transform of a rectangular pulse of τ_CDS_ width—f_3dB1_ = 0.44/τ_CDS1_ = 440 kHz and f_3dB2_ = 0.44/τ_CDS2_ = 880 kHz, respectively [57]. The impedance modules |Z_L_| of the wire introduced by the inductance L at f = f_3dB1_ and f = f_3dB2_ are |Z_L1_| = 2π × f_3dB1_ × L = 0.036 Ω and |Z_L2_| = 2π × f_3dB2_ × L = 0.072 Ω, negligible in comparison with 0.6 Ω of the wire resistance.

The R_s com_ common series resistance 0.6 Ω was introduced by the 25 µm diameter, 1 cm long gold lead wire and its compression bond used to connect the anode electrode with the output pin.

The parallel capacitance C_p_ scales proportionally to the photodiode photosensitive area and was only estimated from measurements of some reverse biased photodiodes, much smaller than the PSD photodiode and made of wafers similar to those used for this PSD. These were time response measurements using fast-decay laser pulses [22,58,59] as well as small-signal reflectance measurements with the vector network analyzer [60]. The C_p_ was estimated to be ca. 1 nF. As for R_p_, such a value for C_p_ was possible due to suppression of the Auger generation with the reverse bias.

The photocurrent i_ph_ at the anode electrode (Figure 6 and Figure 7) is the sum of the photocurrents from all the four cathode electrodes. It can be found for a given input radiation spectral power P_inλ_ and a given spectral current responsivity R_iλ_ of the detector using the relation: i_ph_ = P_inλ_ × R_iλ_. The low frequency R_iλ_ of the PSD photodiode is shown in Figure 10. It was measured with a Fourier transform infrared spectrometer calibrated with a black-body radiation source with an accuracy of +/−20%. Short-wavelength radiation is largely filtered by the wide-gap cathode resistive layer, through which radiation enters the absorber, while the long-wavelength tail of the spectral characteristics is determined by the width and composition of the narrow-gap absorber.

Figure 10 reveals a strong dependence of the responsivity magnitude on V_b_ around 0.16 V, in the range of the high differential resistances, presented in Figure 8. The responsivity, strongly limited by the series resistance, was boosted with V_b_ between 156 mV and 157 mV and then gradually decreased as V_b_ increased. The rise in the responsivity was correlated with the sharp increase in the differential resistance of the detector, where the parallel resistance R_p_ was enhanced in relation to the series resistance. Moreover, at V_b_ in the range of 157 mV to 163 mV, the spectral responsivity exceeded that of the ideal photon counter with the quantum efficiency 70%, conforming to 100% − 70% = 30% of the radiation reflection losses in the detector, mainly at the air-GaAs interface of the chip substrate. This was the evidence of the photoelectric gain g_ph_ greater than unity. The g_ph_ value was estimated to equal ca. 1.7 for V_b_ = 159–160 mV, based on the following formula:(7)$$g_{ph}=\frac{R_{i\lambda }}{e\eta_{r}\eta_{a}/\left(h\frac{c}{\lambda }\right)}$$
where h is Planck’s constant, c is the speed of light in a vacuum, $\left(h\frac{c}{\lambda }\right)$ is the photon energy, η_a_ is the absorption quantum efficiency deduced from fitting of modeling of the infrared absorption and transmission to the measured relative spectral response of the detector and room-temperature transmission of wafers with HgCdTe layers [61], η_r_ = 70% is the quantum efficiency related to the radiation reflection losses, and e is the charge of the electron.

## 6. Pulsed Radiation Measurement

The read-out circuitry (ROC) for the pulsed radiation measurements was realized in the form of a switched integrating transimpedance operational amplifier (op-amp) at each of the PSD’s four outputs. Figure 11 presents a general scheme of the ROC. The integrating op-amp consisted of the fast SiGe-based bipolar input ADA4896 op-amp integrated circuit chip [52], characterized by a low input current and voltage noise density, and the integrating capacitor C_i_ = 220 pF in the op-amp negative feedback loop, thus providing conversion of the integrated photocurrent to the output voltage across the C_i_.

The applied integration intervals were synchronized with 100 ns pulses of input radiation, emitted from the quantum cascade laser (QCL) with the 10 kHz repetition frequency. The timing of integration and sampling of the output voltage was controlled by a TTL-voltage compatible switching. This is illustrated in Figure 12, where Figure 12a shows an example output voltage waveform from one channel with a ca. 0.95 V photo-response to the 100 ns laser pulse and Figure 12b plots the output voltages from all four channels with the unlighted detector. Correlated double sampling (CDS) was applied. The first and second sample of each pair of the correlated samples of the output voltage were measured at time instants M1 and M2, respectively, when the reset switch (RS) was open. The first sample V_M1_ preceded the start of the radiation pulse and followed 0.5 μs after opening the RS at 0 ns. The second sample V_M2_ was taken well after the end of the radiation pulse, when all the photocurrent had been integrated. The voltage V_M1_ was subtracted from the voltage V_M2_. The measurement result was V_M2_−V_M1_, the CDS voltage. Next, the RS was closed for a time more than sufficient for discharge of the integrating capacitor. However, in the reset state (zeroing), the voltage across the capacitor settled at a level slightly different from zero. This zeroing offset could have been due to the op-amp input offset voltage, the flow of the uncompensated bias current through the RS, and charge injection during change of the RS state. The CDS dramatically reduced 1/f noise and the zeroing offset. The remaining output offset was due to the inaccurate compensation of the dark current, a small part of which flowed through the integrating capacitor.

This switched (also called box-car) integration of a signal improves the signal-to-noise ratio and is the preferred method for measuring weak pulsed radiation if an increase in measurement time is tolerated, such as when measuring pulse energy.

The input impedance of the ROC should be minimized so as not to unduly limit the photocurrent and the linearity of the PSD. This was achieved with the fast transimpedance op-amps, characterized by the relatively high open loop gain crossover frequency. The impedance mismatch was not significant here due to the limited bandwidth imposed by the applied value of the integration time τ_CDS_.

The ROC should ensure the constant reverse bias voltage across the photodiode along with compensation of its bias (dark) current. The bias voltage should be within a relatively narrow range (Figure 8 and Figure 10), allowing sufficient depletion of carriers throughout the absorber, thus increasing the parallel impedance in relation to the series impedance so that the latter does not unreasonably limit the responsivity and linearity. The optimal bias voltage is then the same for all the channels. Its common regulation, as shown in Figure 11, was correct, as opposed to being separate for each channel.

Two detectors, each with a separate PSD chip from the same wafer, were assembled and integrated with the read-out electronics, forming two compact samples of PSD modules #1 and #2. The modules were attached to a heat-sink, which helped in the electronic stabilization of the detector operation temperature with a precision of 0.01 K. The output voltage was measured with an oscilloscope and mapped across the photosensitive area in a setup shown in Figure 13.

The laser beam was focused with a parabolic mirror onto the photosensitive area of the PSD. A motorized X-Y stage, actuated by stepper motors, enabled scanning of the photosensitive area with the radiation spot. The measurement was controlled by a script written in Python programming language, installed on a computer (PC). The script collected and organized the sampling data of the output voltages U_A_, U_B_, U_C_, and U_D_ from all four channels, obtained from the oscilloscope, and the data of the actual (X, Y) position of the radiation spot on the photosensitive area, received from the encoders of the stepper motors through their controllers. The actuation of the stepper motors was synchronized by the script with the time the PC received data from the oscilloscope and the encoders. Based on the gathered data, the script calculated the beam spot position (X_PSD_, Y_PSD_) using Formulas (1) and (2), assigned it to the actual (X, Y) position data from the encoders, and presented the results in a graphical form.

An example waveform of the output voltage together with the input radiation power from the pulsed laser is presented in Figure 14. The radiation power was measured using another, uncooled photovoltaic detector type PVM-10.6-4×4, manufactured by VIGO Photonics S.A. (VIGO) (Ożarów Mazowiecki, Poland) [16,17,62], with the HgCdTe heterostructure consisting of multiple photovoltaic junctions connected in series that are distributed over a large photosensitive area of 4 mm × 4 mm. The PVM-10.6-4×4 detector is characterized by a large linearity range of the input radiation power and a low time constant of 1.0–1.5 ns, compared to the PSD presented herein.

Based on the PSD equivalent circuit from Figure 7 and the values of its elements given therein, the time constant τ_PSD_ of the single-channel photoresponse can be approximated as follows:τ_PSD_ = [R_p_‖(2 × R_scom_ + 0.5 × R_sh_ + Z_in amp_)] × C_p_ ≈ (2 × R_scom_ + 0.5 × R_sh_ + R_in amp_) × C_p_ = (1.2 Ω + 0.5 × 5.5 Ω + 8 Ω) × 1 nF = 12 ns(8)
where the detector is loaded by the integrating preamplifier input impedance Z_in amp_ (Figure 11), which in fact is the resistance R_in amp_ for the considered frequencies > 100 kHz and can be calculated as follows:Zin amp = Rin amp = (2πCifc)^−1^ = (2π × 220 pF × 90 MHz)^−1^ = 8 Ω(9)
where f_c_ is the open loop gain crossover frequency of the ADA4896 op-amp used [52].

As was expected, neither the PSD nor the preamplifier input contributed significantly to the time of the PSD response to the 100 ns laser pulse (Figure 14).

Figure 15 demonstrates the characteristics of the signal from the PSD module output U_out_ = U_A_ ≈ U_B_ ≈ U_C_ ≈ U_D_ versus the energy of the radiation pulse of width t_pulse_ = 100 ns and shows the linear operation limit equal to U_out_3%dev_ (t_pulse_ = 100 ns) = 1.5 V at a 3% drop from the linear response. This corresponds to 0.54 nJ +/− 20% of the single-pulse energy at center illumination, where the signals from each channel are approximately equal, within +/−10% of the average. With deviation from center illumination, the pulse energy corresponding to the linear operation limit decreases to compensate for the increasing signal at the output with the highest voltage. The energy of the radiation pulse was measured by integrating the radiation power waveform obtained with the VIGO PVM-10.6-4×4 detector (Figure 14).

It is optimal to operate within and not too far from the detector’s linear operation limit. Exceeding this limit could lower the differential impedance of the detector, the high value of which is crucial for good performance and stable operation. However, assume this PSD operates at the signal close to its linear operation limit, at a 3% drop from the linear response. Then, the photocurrent per channel I_ph/ch_3%dev_ = 0.8 mA:I_ph/ch_3%dev_ = U_out_3%dev_ × C_i_/K_u_/t_pulse_ = 1.5 V × 220 pF/4/100 ns = 0.8 mA(10)
where K_u_= 4 V/V is the gain of the output stage of the preamplifier (Figure 11). This photocurrent puts a significant voltage U_s_3%dev_ = I_ph/ch_3%dev_ × R_s_ = 10 mV across the series resistance R_s_ = 2 × R_scom_ + 0.5 × R_sh_ + R_in amp_ = 1.2 Ω + 0.5 × 5.5 Ω + 8 Ω = 12 Ω, from Equation (8), thus reducing by 10 mV the reverse bias voltage across the absorber and its interfaces. With a constant input radiation power, the photocurrent and photovoltage put across the series resistance of the corresponding channel increase with the distance of the radiation beam from the central illumination point. Thus, the reverse bias voltage, if set to optimal for central illumination, will be less than optimal the farther the radiation beam is moved away from the center. This also reduces the detector resistance in relation to the series resistance, lowers the responsivity and adds to the position nonlinearity.

The PSD area was scanned with 100 ns pulses of QCL radiation of λ = 10.53 µm and energy of 0.26 nJ, focused on a spot with a Gaussian-like profile (Figure 16) of the 1/e^2^ diameter 240 µm, with a repetition rate of 10 kHz. The PSD was operated at 205 K, and the reverse bias voltage V_b_ = 0.16 V, using τ_CDS_ = 1 µs. The resulting signal chromatic maps of the photosensitive area are presented in Figure 17, Figure 18 and Figure 19. The scan results in Figure 20 and Figure 21 were obtained under the same conditions.

In Figure 17, the map of the sum of all four outputs shows the periphery with > 30% of the signal attenuation, covering about half of the photosensitive area. This may be caused by the radiation spot extending partially outside the photosensitive area. In addition, other nonuniformities are clearly visible, also on subsequent maps in Figure 18 and Figure 19, taken for the signal from each PSD channel separately. No anti-etaloning solutions were applied, so etaloning due to radiation interferences in the wide-gap layers outside the absorber might contribute here. Inhomogeneities of the material layers, particularly the resistive layer, the cathode electrode metal–semiconductor contact, and the absorber and its interfaces may also play a role. Slightly more uniform maps were obtained for the next sample of the PSD module #2 (Figure 19).

Figure 20 plots linear scan results as the difference between the signals from the opposite channels versus the radiation spot’s actual position given by the encoders. Also shown is the interesting known linearization effect when the difference between the signals is divided by their sum. The linearization hides the nonuniformities visible in the plots of the simple difference (Figure 20a).

It was also interesting to measure a map of the difference, or error, between the (X_PSD_, Y_PSD_) spot position determined from the PSD signals, using Equations (1) and (2), and the actual (X, Y) position given by the encoders. This is shown in Figure 21. The noise was merely visible due to the 32 averaging no. used. Nonlinearities are clearly visible, with barrel-like shapes characteristic of the tetra-lateral configuration of the cathode electrodes [13].

## 7. Noise Characterization

Figure 22 illustrates primary noise contributions for a typical connection of the PSD and the ROC. $\bar{u_{nA}}$, $\bar{u_{nB}}$, $\bar{u_{nC}}$, and $\bar{u_{nD}}$ at the respective outputs are the roots of the sum of the signal variances from uncorrelated noise sources.

The output noise is estimated under the following assumptions: the dark condition; the detector dark current is spread evenly between the four cathode electrodes A, B, C, and D, with the dark current per channel I_d/ch_ = 12 mA at V_b_ = 0.16 V; and the voltage and current rms noises of the preamplifier input, $\bar{u_{na}}=1\;nV/\sqrt{Hz}$ and $\bar{i_{na}}=2.8\;pA/\sqrt{Hz}$, respectively, are the same for all the preamplifiers used. Then, the same formula may be used for the total rms of the fluctuations $\bar{Q_{nA}}$, $\bar{Q_{nB}}$, $\bar{Q_{nC}}$, and $\bar{Q_{nD}}$ of the charges Q_A_, Q_B_, Q_C_, and Q_D_, respectively, integrated during the CDS time interval τ_CDS_ on the capacitance C_i_ = 220 pF of the respective channels, and for the resulting noises $\bar{u_{nA}}$, $\bar{u_{nB}}$, $\bar{u_{nC}}$, and $\bar{u_{nD}}$:(11)$$\bar{Q_{nA}}=\sqrt{\tau_{CDS}\int_{-\infty }^{+\infty }\left(2{eI}_{d/ch}g_{ph}\left(1+f_{k}^{1.1}/f^{1.1}\right)/\left(1+f_{l}^{1.1}/f^{1.1}\right)+4{kT}_{d}/R_{\mathrm{sh}}+{\bar{u_{na}}}^{2}/{\left|Z_{d}\right|}^{2}+{\bar{i_{na}}}^{2}\right){sinc}^{2}\left(\pi f\tau_{CDS}\right)df+{kT}_{e}C_{i}}$$
where g_ph_ = 1.7 is estimated according to Equation (7); f_k_ is the 1/f noise higher crossover (knee) frequency, below which the 1/f noise dominates; f_k_ = 1 MHz is fit to the measured PSD photodiode noise current spectral density $\bar{i_{nd}}$, shown in Figure 23, where $0.5\times \bar{i_{nd}}=\bar{i_{nd/ch}}$; the 1.1 exponent at the f_k_ and the frequency variable f is a result of fit to the measured $\bar{i_{nd}}$ in the range, where the 1/f noise dominates; several f_l_ values are assumed, where f_l_ is the 1/f noise lower crossover frequency related to the time of the measurement [63], i.e., to τ_CDS_; T_d_ = 205 K; T_e_ = 295 K is the operating temperature of the preamplifier; Z_d_ = R_sh_/2 + [(j2πfC_p_)^−1^‖R_p_]; k is the Boltzmann constant; and e is the charge of the electron.

The high C_b_=100 µF capacity connected in parallel with the V_b_ voltage source with the internal resistance R_b_ = 50 Ω limits the noise bandwidth at the output of the V_b_ voltage source to only 1/(4C_b_R_b_) = 50 Hz. This makes the noise of this source negligible, so it is not included in Equation (11).

The output noise voltage is calculated as follows:(12)$$\bar{u_{nA}}=\bar{u_{nB}}=\bar{u_{nC}}=\bar{u_{nD}}=\sqrt{{\left(\frac{\bar{Q_{nA}}}{C_{i}}{\times K}_{u}\right)}^{2}+\bar{u_{nOSC}^{2}}}$$
where K_u_= 4 V/V is the gain of the output stage of the preamplifier (Figure 11 and Figure 22) and $\bar{u_{nOSC}}$ is the vertical (voltage) rms noise of the oscilloscope for a given volt/div setting applied.

There could be some negative correlation between the respective noise contributions in the u_nA_, u_nB_, u_nC_, and u_nD_ output noises originating from thermal (Johnson) noise of the resistive layer. This correlation is expected to increase with the PSD photodiode parallel impedance Z_p_ = R_p_‖(j2πfC_p_)^−1^. However, the contribution of the Johnson noise of the resistive layer is insignificant due to the limited noise bandwidth imposed by the CDS integrating interval τ_CDS_ through the sinc(πfτ_CDS_) transfer function and because of domination of the huge 1/f noise of the PSD photodiode. This is shown in Figure 23, demonstrating several contributions to the noise current spectral density on the integrating capacitor. The black line fits points of half of the measured PSD photodiode noise current spectral density, $\bar{i_{nd/ch}}=0.5\times \bar{i_{nd}}$, obtained with all the cathode outputs connected together, before integration of the PSD with electronics, and the red line provides the same noise with the added calculated Johnson noise current spectral density of the resistive layer. The contribution of the preamplifier input voltage noise spectral density $\bar{u_{na}}$, i.e., $\bar{u_{na}}/\left|Z_{d}\right|$ noise current spectral density, is insignificant too, for the same reason.

In addition, $\bar{i_{na}}=2.8\;pA/\sqrt{Hz}$, $\bar{u_{nOSC}}=165\;µV$, and $\sqrt{{kT}_{e}C_{i}}/C_{i}\times K_{u}=17\;µV$ are negligible.

The blue line in Figure 23 demonstrates strong limitation to the noise by the 1/f noise lower crossover frequency f_l_ assumed to equal 1/(2πτ_CDS_), as in [64]. Following [63], f_l_ should be in inverse proportion to the measurement time, equal to τ_CDS_ here, and the value of f_l_ can be obtained from measurements or a theory. The output noise voltage estimated by Equations (11) and (12) for the integration time τ_CDS_ = 1 µs and 0.5 µs and for the several values of f_l_ is given in Figure 24.

The output noise voltage measurement was carried out using the ROC and PSD connections from Figure 11 and a fast oscilloscope operated at 10 GigaSamples/second and 500 MHz bandwidth. The CDS technique described in Section 6 was used and no laser light was applied, so no photosignal was produced. Figure 25 is a histogram of the CDS voltages measured at τ_CDS_ = 1 μs and with the PSD operated at V_b_ = 0.16 V and T_d_ = 205 K. The Gaussian-like distributions of the CDS voltage with its rms noises $\bar{u_{nA}}=3.6\;mV$, $\bar{u_{nB}}=3.4\;mV$, $\bar{u_{nC}}=3.5\;mV$, and $\bar{u_{nD}}=3.6\;mV$ at the respective outputs were obtained. The measured values for the rms noise were 1.6 times greater than the output noise voltage estimated by Equations (11) and (12) for the same τ_CDS_ and conditions of PSD operation if f_l_ = 1/(2π τ_CDS_) was assumed, as in [64]. This is still reasonably good consistency between the measured and estimated noise values.

Finally, the measurement resolution of the (X_PSD_, Y_PSD_) position was determined, $\bar{x_{nPSD}}$ and $\bar{y_{nPSD}}$, based on:Equations (3) and (4);The measured noise of the output voltage;The output signals measured for the positions of the radiation spot where the maximum difference between the signals from the opposite cathode electrodes occurred, i.e., beyond the central area where a similar or better resolution was achieved.

The data are summarized in Table 3. The position resolution of 3–6 µm without averaging was obtained using the τ_CDS_ interval of 1 µs and 0.26 nJ 100 ns radiation pulses. However, the photocurrent was actually integrated just during the 100 ns radiation pulse. For the radiation pulse wider than the applied CDS interval, the photocurrent was integrated 10 times longer, over the entire interval τ_CDS_ = 1 µs, and the signal-to-noise ratio also increased 10 times. Thus, the position resolution of this PSD is 10 times better, i.e., 0.3–0.6 µm.

This position resolution could be further improved with modest averaging. Averaging improves signal-to-noise ratio N^1/2^ times, where N is the averaging number. Thus, the resolution is improved N^1/2^ times, too, according to Equations (3) and (4). The measurement time equal to τ_CDS_ = 1 µs plus a few microseconds for resetting would be increased N times then.

Using Equation (6), taking the dark current noise density per one channel $\bar{i_{nd/ch}(1\;kHz)}$ = 4.1 × 10^−9^ A/Hz^1/2^ (Figure 23) and the dark current per one channel at the applied reverse bias I_d/ch_ = 12 mA (Figure 8), we obtain the 1/f noise related Tobin coefficient:(13)$$\alpha =4.1\times 10^{-9}\frac{A}{\sqrt{Hz}}\times \frac{\sqrt{1000\;Hz}}{0.012A}=1.1\times 10^{-5}$$
which is a relatively small value compared to some other data from the last few years reported for HgCdTe devices at higher operating temperatures, e.g., for an MOCVD grown MWIR barrier detector at 205 K—α = 2.7 × 10^−4^ [54] and for a liquid phase epitaxy (LPE) grown MWIR photodiode at 290 K—α = 2–40 × 10^−5^ [65]. However, this may reflect not so much the quality of our photodiode as the larger share of the diffusion current in its dark current. Referring to diffusion-current-limited LPE MWIR and LWIR photodiodes at temperatures ≥ 200 K [66], α ≈ 2 × 10^−6^ can be found.

However, dislocations intercepting the junction are likely to be a challenge, limiting the production yield of our relatively large 1 × 1 mm^2^ photodiodes. The probability of a dislocation intercepting the junction is proportional to the dislocation density and the junction area. We obtained the 1 mm × 1 mm photodiodes with yields a few times lower compared to devices with a smaller photosensitive area of 100 µm × 100 µm. The defective detectors had high dark currents and too-low resistances.

## 8. Summary and Conclusions

The lateral-effect position-sensitive Peltier-cooled LWIR detector was recently demonstrated [20] for the first time to the authors’ knowledge and described in more detail here. It was based on a modified PIN HgCdTe photodiode with a spectral sensitivity range of 3–11 μm, forming the tetra-lateral PSD with a photosensitive area of 1 × 1 mm^2^. The device was coupled to integrating preamplifiers and scanned with a focused radiation beam. It was capable of achieving a position resolution of 0.3–0.6 µm using a box-car integration time of 1 µs with correlated double sampling, which dramatically reduced the noise bandwidth and limited the contribution of the huge 1/f noise of this photodiode and the offsets of the output signal.

In order to improve the position linearity of this PSD, the cause of the observed nonuniformities in the signal maps should be further clarified, e.g., by comparison with maps obtained with a thermal radiation source. Anti-etaloning solutions should be considered. Inhomogeneities of the on-chip material layers, particularly the resistive layer, can be overcome by improving chip processing. However, the input impedance of integrating op-amps is comparable to and greater than the resistive layer sheet resistance, limiting the PSD performance, and differences between op-amp chips may also play a role. To avoid this, it is required to increase the impedance of the photodiode and the resistance of the resistive layer accordingly.

Due to the low yield and higher-than-expected thermal G-R, TAT, and resulting 1/f noise of this PSD, our technology should be investigated for improving dislocation density and surface passivation.

A 20-fold decrease in the dark current compared to the level exhibited by this PSD, possible with the fully depleted absorber, could significantly reduce the 1/f noise. Another way to avoid or lessen the 1/f noise is to minimize the depletion volume, e.g., with 0 V biased inter-band cascade infrared photodetectors based on InAs/InAsSb superlattices [50] or biased barrier devices, such as nBn made of HgCdTe [51], both optimized for suitably high resistance and fast response.

Resetting the integrating capacitor extends the time between position measurements to a few microseconds. With sufficiently low noise, non-integrating op-amps can be used that do not require resetting between measurements, allowing full use of the small time constant of this PSD, provided that the lead wire inductance is lowered according to the increased operating frequency.

## Figures and Tables

**Figure 1 sensors-23-04915-f001:**
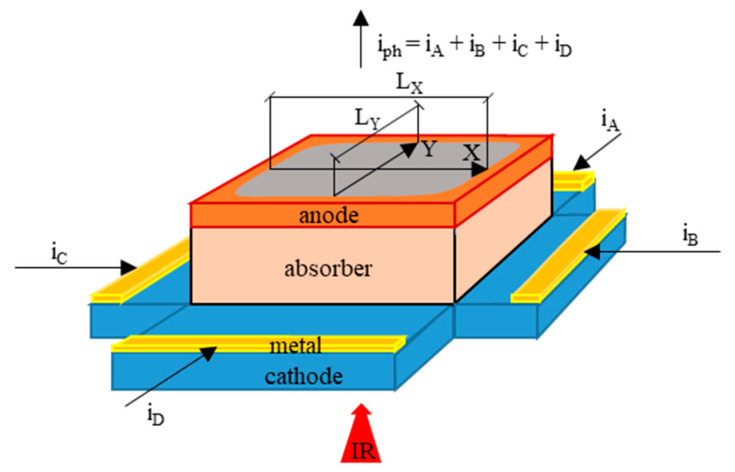
Principle of operation of the PSD tetra-lateral photodiode (adapted with permission from Ref. [20]; copyright: 2022, SPIE). The photosensitive area is in grey. i_A_, i_B_, i_C_, and i_D_ are the photocurrents flowing to the respective preamplifiers of channels A, B, C, and D.

**Figure 2 sensors-23-04915-f002:**
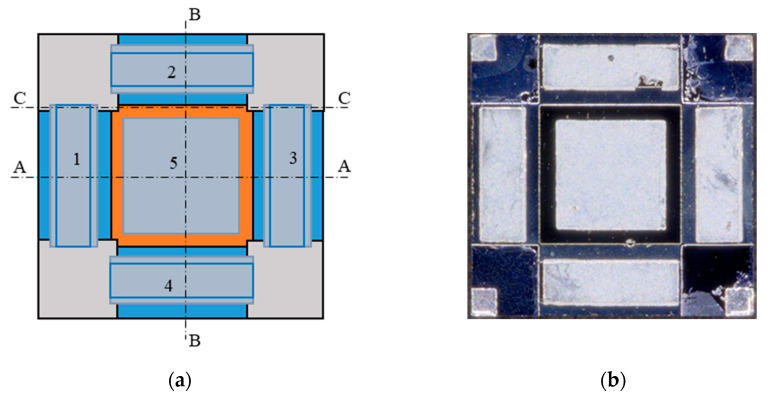
TL-PSD chip—top view: (**a**) a schematic drawing: 1, 2, 3, and 4 denote indium cathode electrodes, and 5 denotes an indium anode electrode; cross-sections A–A, B–B, and C–C are shown in Figure 3 and Figure 4; (**b**) a micrograph of the fabricated chip.

**Figure 3 sensors-23-04915-f003:**
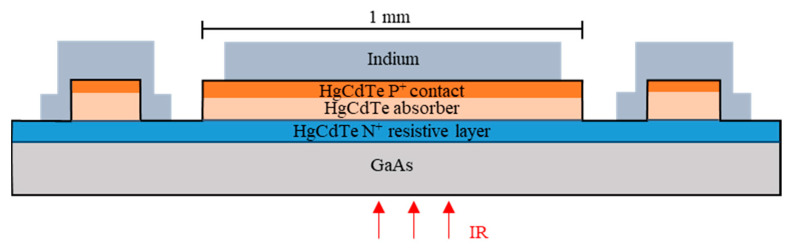
TL-PSD chip—the A–A and B–B cross-section, according to Figure 2.

**Figure 4 sensors-23-04915-f004:**
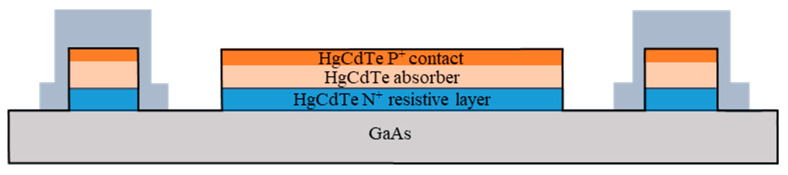
TL-PSD chip—the C–C cross-section, according to Figure 2.

**Figure 5 sensors-23-04915-f005:**
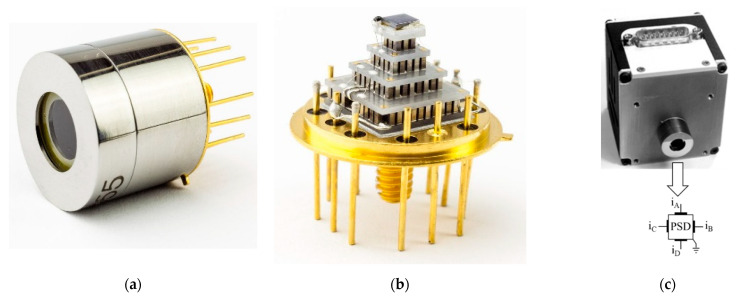
Detector package (adapted with permission from Ref. [20]; copyright: 2022, SPIE): (**a**) the hermetic housing of the TL-PSD chip; (**b**) the TL-PSD chip on the four-stage Peltier cooler cold finger; (**c**) the TL-PSD detector integrated with the electronics package, and a spatial orientation of the cathode electrodes of the signal channels A, B, C, and D is also shown.

**Figure 6 sensors-23-04915-f006:**
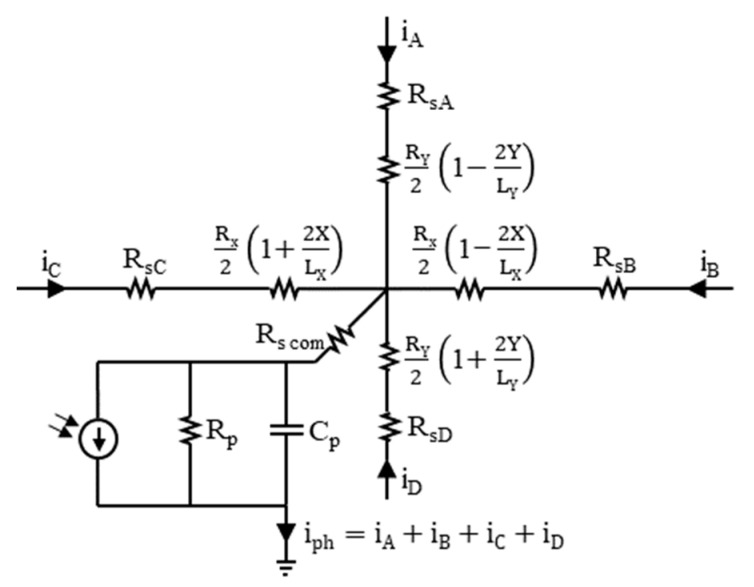
TL-PSD equivalent circuit with the approximated formulas for the resistances of the photocurrent divider in the resistive layer, related to the light spot position (X, Y), where (0, 0) is in the center of the photosensitive area. The following values were determined at the detector operating temperature T_d_ = 205 K and reverse bias V_b_ ≈ 0.16 V: R_p_ ≥ 100 Ω; C_p_ ≈ 1 nF; R_s com_ ≈ 0.6 Ω—the lead wire resistance; R_X_ and R_Y_ may be assumed to equal R_sh_ = 5.5 Ω—the measured sheet resistance of the resistive layer; R_sA_ = 8.9 Ω, R_sB_ = 8.9 Ω, R_sC_ = 9.0 Ω, and R_sD_ = 9.2 Ω, with the main contribution of 8 Ω of the preamplifier input impedance, later calculated using Equation (9), and with minor contributions of contact resistance values from Table 2 and 0.6 Ω of the lead wire resistance.

**Figure 7 sensors-23-04915-f007:**
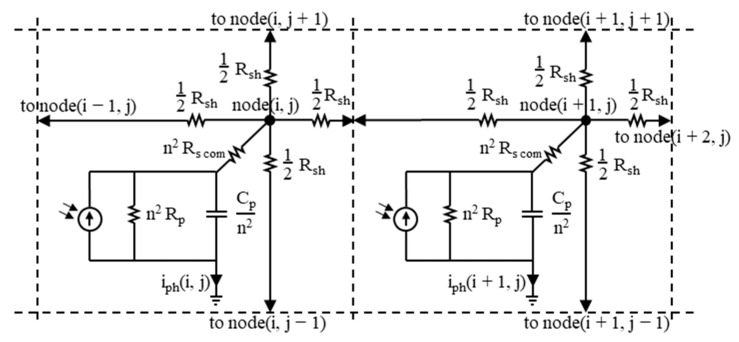
Two cells of the TL-PSD equivalent circuit suited for the finite element analysis. The following values were determined at T_d_ = 205 K and V_b_ ≈ 0.16 V: R_p_ ≥ 100 Ω; C_p_ ≈ 1 nF; R_sh_ = 5.5 Ω—the resistive layer sheet resistance; R_s com_ = 0.6 Ω—the lead wire resistance.

**Figure 8 sensors-23-04915-f008:**
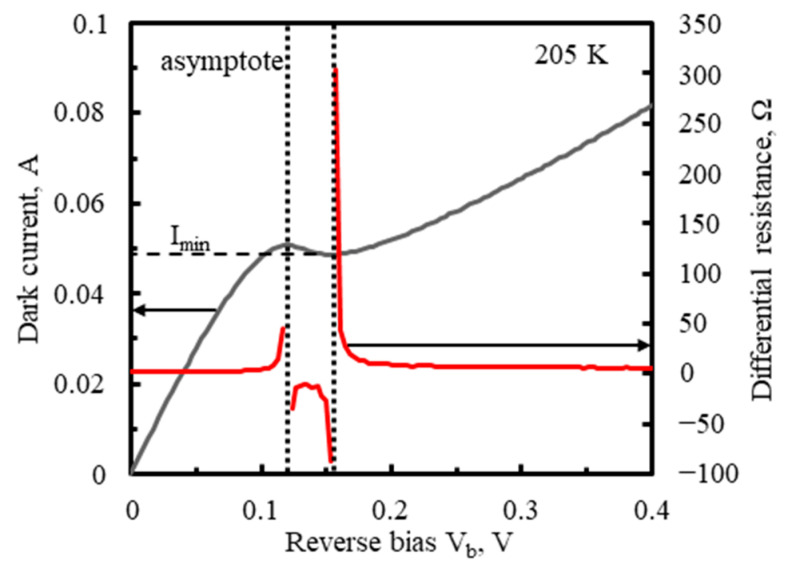
Dark I−V characteristics of the TL−PSD photodiode with the 10–11 µm long wavelength cut-off at T_d_ = 205 K.

**Figure 9 sensors-23-04915-f009:**
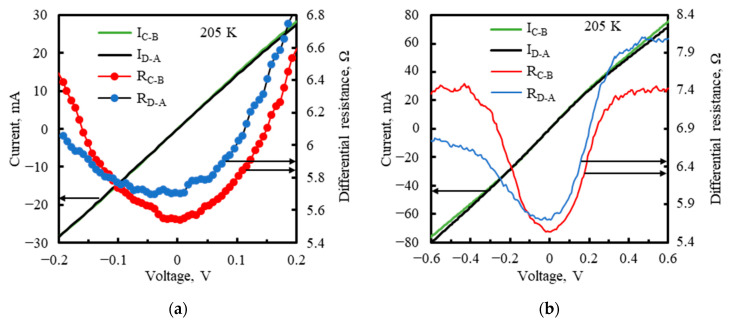
Current and differential resistance versus voltage between the opposite cathode electrodes. The DC measurement at T_d_ = 205 K with the photodiode electrically open. R_C-B_ denotes resistance between the electrodes C and B, and R_D-A_ denotes resistance between the electrodes D and A. These are resistances measured between the respective output pins outside the PSD hermetic package minus the resistance 1.2 Ω of two thin 25 µm diameter golden lead wires, connecting the output pins with the cathode electrodes. (**a**) Results shown in the bias voltage range −0.2 V to +0.2 V. (**b**) The same, with added results in the greater bias range.

**Figure 10 sensors-23-04915-f010:**
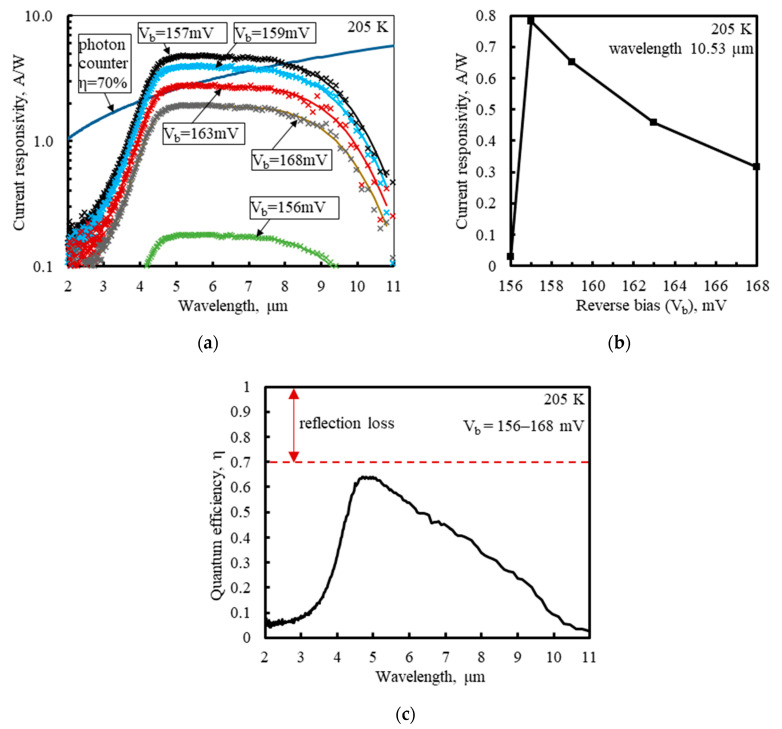
Spectral responsivity of the PSD photodiode at T_d_ = 205 K, from the photocurrent at the anode common contact, measured with the Fourier transform infrared spectrometer for the various reverse bias voltages V_b_: (**a**) spectral characteristics (adapted with permission from Ref. [20]; copyright: 2022, SPIE); (**b**) rapid change in the responsivity with V_b_ around 0.16 V; (**c**) spectral quantum efficiency η = η_r_ × η_a_.

**Figure 11 sensors-23-04915-f011:**
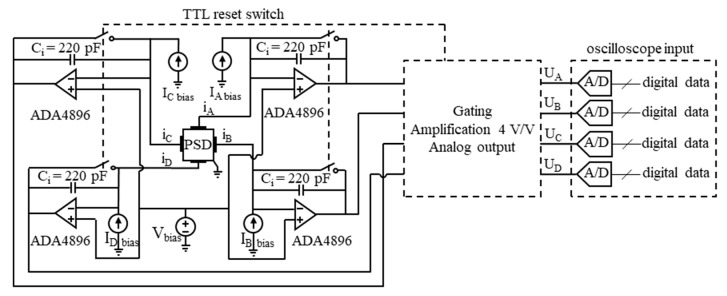
General scheme of the read-out electronic circuit suited to the PSD photodetector, with the switched integrating preamplifiers, current sources compensating the reverse bias current individually for each channel, and one voltage source ensuring the constant reverse bias voltage for all the four channels. The output voltage was measured with a sampling analog-to-digital converter of the oscilloscope input.

**Figure 12 sensors-23-04915-f012:**
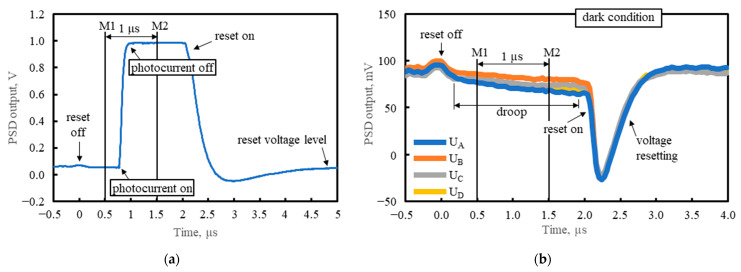
Reset-and-integrate timing with the sampling time instants M1 and M2 shown: (**a**) example waveform of the output voltage from one channel, when the detector was lighted with the 100 ns laser pulse; (**b**) waveforms of the output voltages from all the channels, A, B, C, and D, when the detector was shielded from radiation.

**Figure 13 sensors-23-04915-f013:**
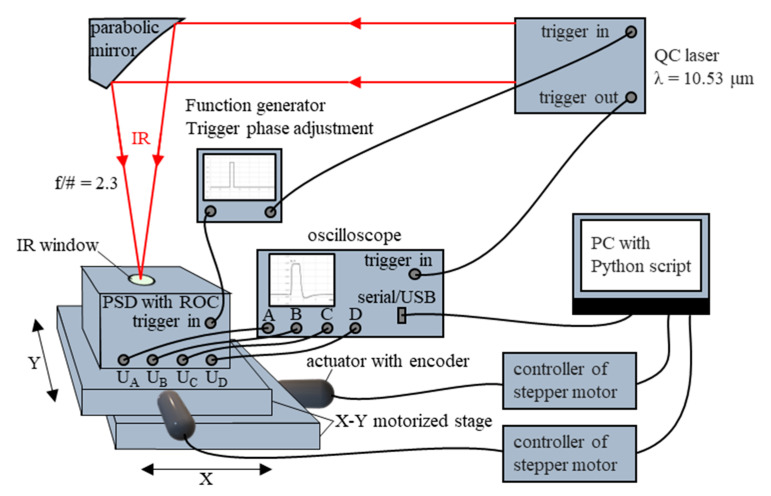
Measurement setup for mapping a PSD photosignal across the photosensitive area.

**Figure 14 sensors-23-04915-f014:**
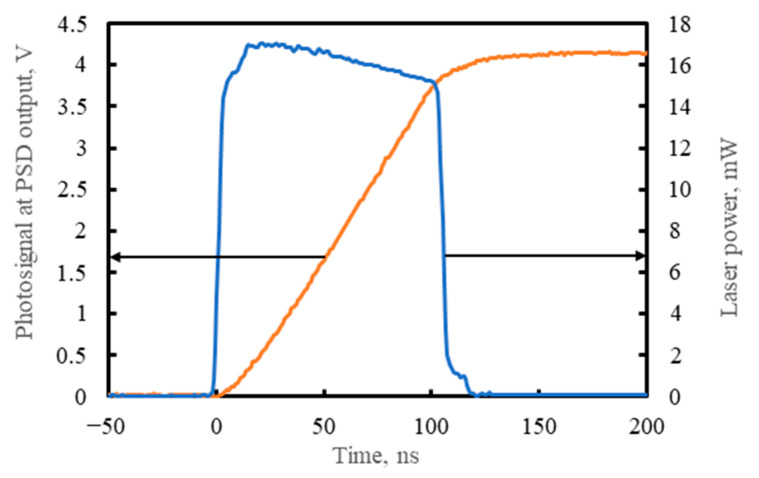
Example waveforms: the input power of QCL radiation focused centrally on the PSD photosensitive area and the voltage signal from one of the PSD module outputs (adapted with permission from Ref. [20]; copyright: 2022, SPIE).

**Figure 15 sensors-23-04915-f015:**
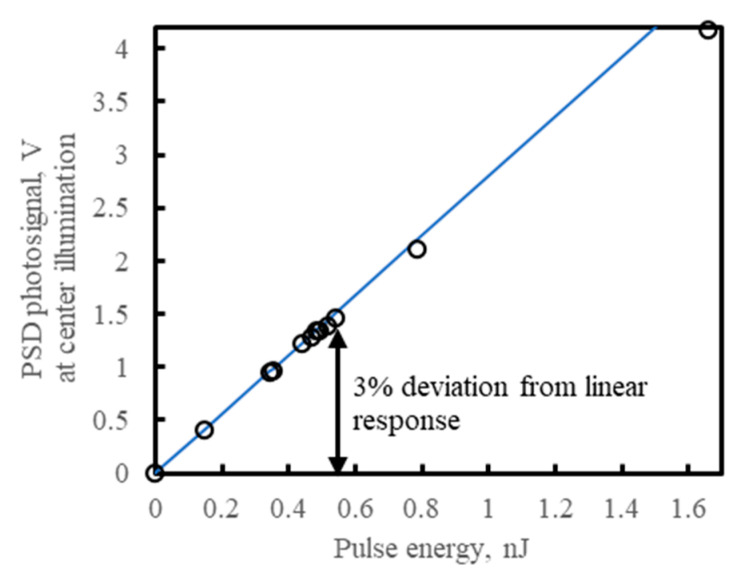
Linearity characteristics—the PSD module output signal versus the energy of the single radiation pulse at center illumination, similar for the samples #1 and #2 of the PSD modules. Averaging no. 64. The QCL 100 ns pulses λ = 10.53 µm, the repetition rate 10 kHz, the Gaussian-like radiation spot of the 1/e^2^ diameter 240 µm, τ_CDS_ = 1 µs. The PSD photodiode was operated at T_d_ = 205 K and the reverse bias V_b_ = 0.16 V.

**Figure 16 sensors-23-04915-f016:**
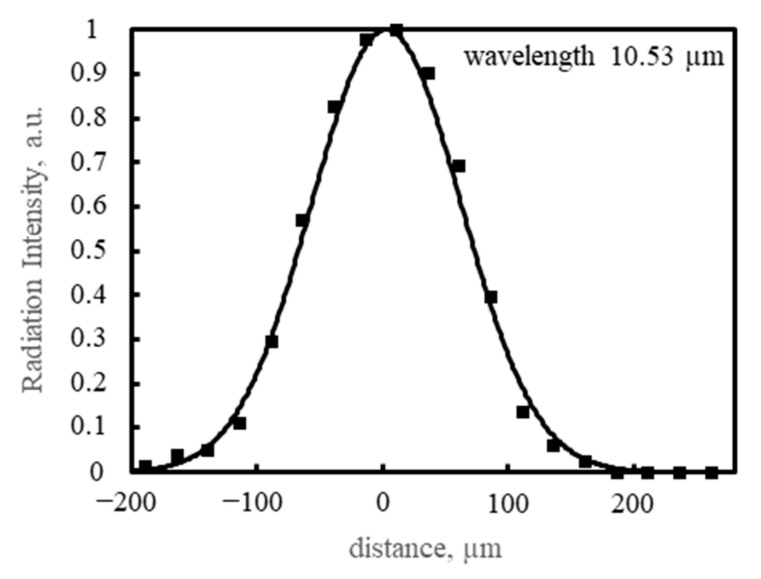
Gaussian-like profile of a radiation spot emitted by the QCL. The 1/e^2^ diameter is 240 µm, measured with another Peltier-cooled HgCdTe photodiode with a small 30 µm × 30 µm photosensitive area, type PV-3TE-10.6-0.03 × 0.03, manufactured by VIGO Photonics S.A.

**Figure 17 sensors-23-04915-f017:**
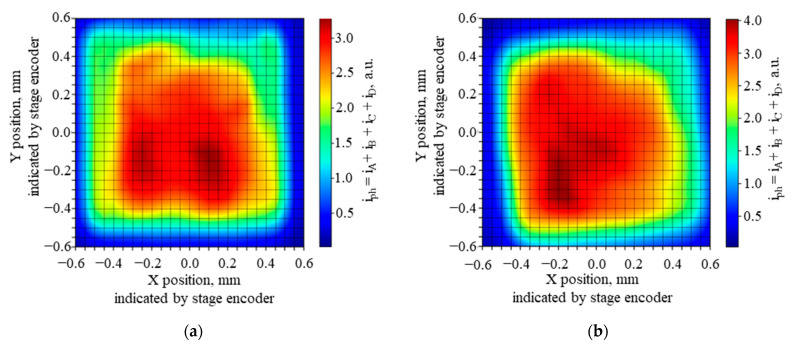
Chromatic map of the sum of the signals from the channels A, B, C, and D for two samples of the PSD modules: (**a**) sample #1; (**b**) sample #2. Averaging no. 32. The maximal signal from a single channel was 1.2 V.

**Figure 18 sensors-23-04915-f018:**
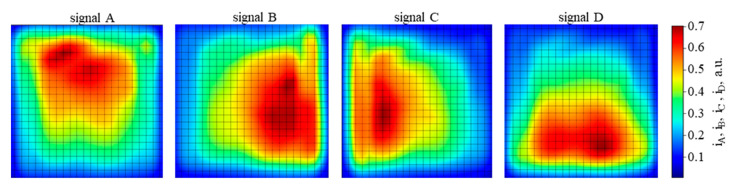
Signal chromatic maps of the photosensitive area of the PSD module sample #1, taken from the individual channels separately. Averaging no. 32. The maximal signal 1.2 V.

**Figure 19 sensors-23-04915-f019:**
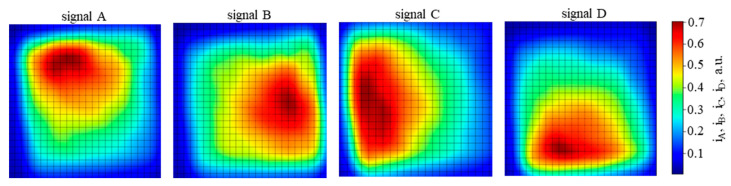
Signal chromatic maps of the photosensitive area of the PSD module sample #2, taken from the individual channels separately. Averaging no. 32. The maximal signal 1.2 V.

**Figure 20 sensors-23-04915-f020:**
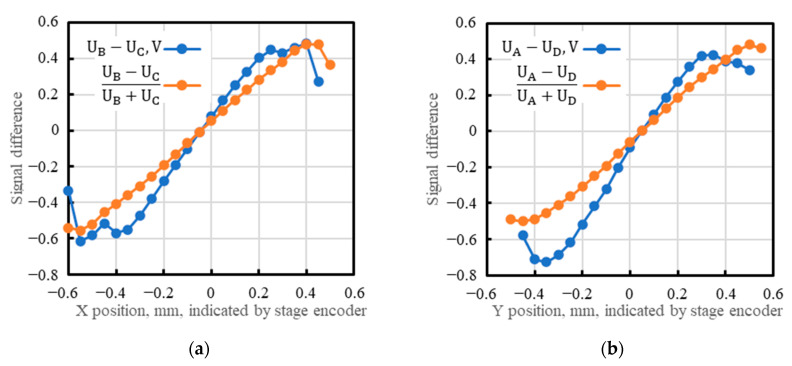
Difference between signals from the opposite channels versus the radiation spot’s actual position: (**a**) U_B_ − U_C_ and (U_B_ − U_C_)/(U_B_ + U_C_) versus the position along the X axis; (**b**) U_A_ − U_D_ and (U_A_ − U_D_)/(U_A_ + U_D_) versus the position along the Y axis. (X = 0, Y = 0) is the middle of the photosensitive area.

**Figure 21 sensors-23-04915-f021:**
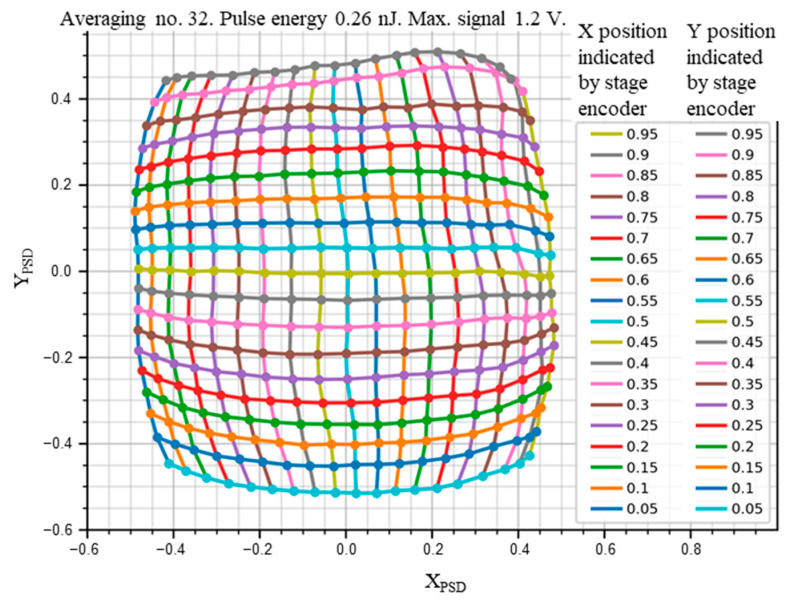
PSD position error chart (adapted with permission from Ref. [20]; copyright: 2022, SPIE). Colored curves with the (X_PSD_, Y_PSD_) points are assigned by color to the respective actual position (X, Y) values, given in two columns at the right. The encoders were zeroed at the corner of the area, so the central point is (X = 0.5, Y = 0.5), corresponding to (X_PSD_ = 0, Y_PSD_ = 0). The PSD module sample #1.

**Figure 22 sensors-23-04915-f022:**
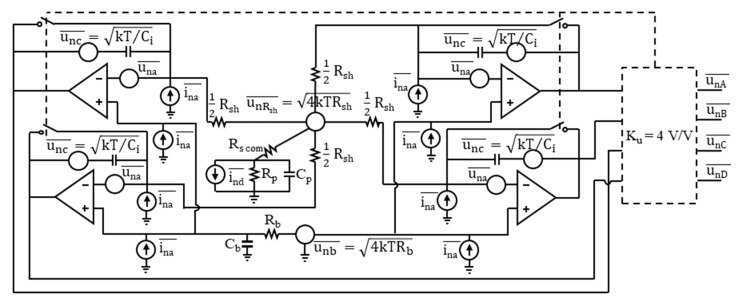
Noise sources, rms, based on the connection diagram shown in Figure 11 and the detector’s equivalent circuit shown in Figure 7: $\bar{u_{na}}$ and $\bar{i_{na}}$ are the voltage and current, respectively, noise spectral densities of the preamplifier input; $\bar{u_{nb}}$ and $\bar{u_{nR_{\mathrm{sh}}}}$ are the noise voltage spectral densities of the bias voltage source and the resistive layer sheet resistance, respectively; $\bar{u_{nc}}$ is the noise voltage of the thermal fluctuations of the charge left on the integrating capacitor after opening an ideal switch.

**Figure 23 sensors-23-04915-f023:**
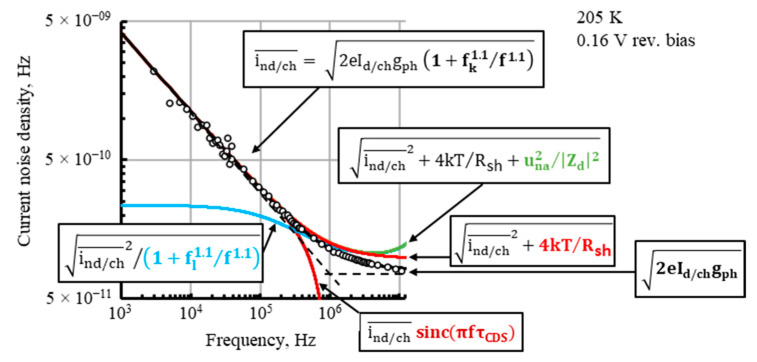
Contributors to the noise current spectral density on the integrating capacitor of the PSD module. The measured dark current noise density per one channel $\bar{i_{nd/ch}}$ is marked in black. Other respective highlighted expressions were calculated and taken from Equation (11). Components with sinc(πfτ_CDS_) and f_l_ factors are shown for τ_CDS_ = 1 µs, where f_l_ is assumed to equal 1/(2πτ_CDS_).

**Figure 24 sensors-23-04915-f024:**
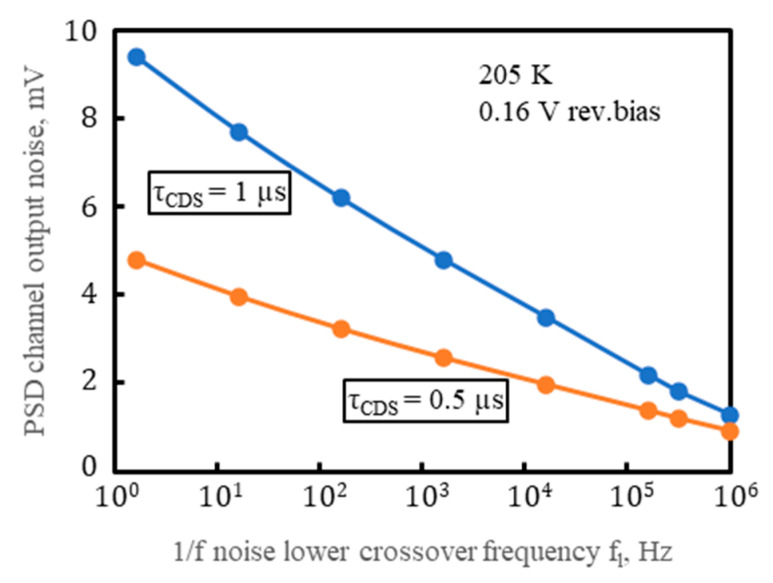
The output noise voltage estimated by Equations (11) and (12) for the integration time τ_CDS_ = 1 µs and 0.5 µs and for the several values of the 1/f noise lower crossover frequency f_l_ (adapted with permission from Ref. [20]; copyright: 2022, SPIE).

**Figure 25 sensors-23-04915-f025:**
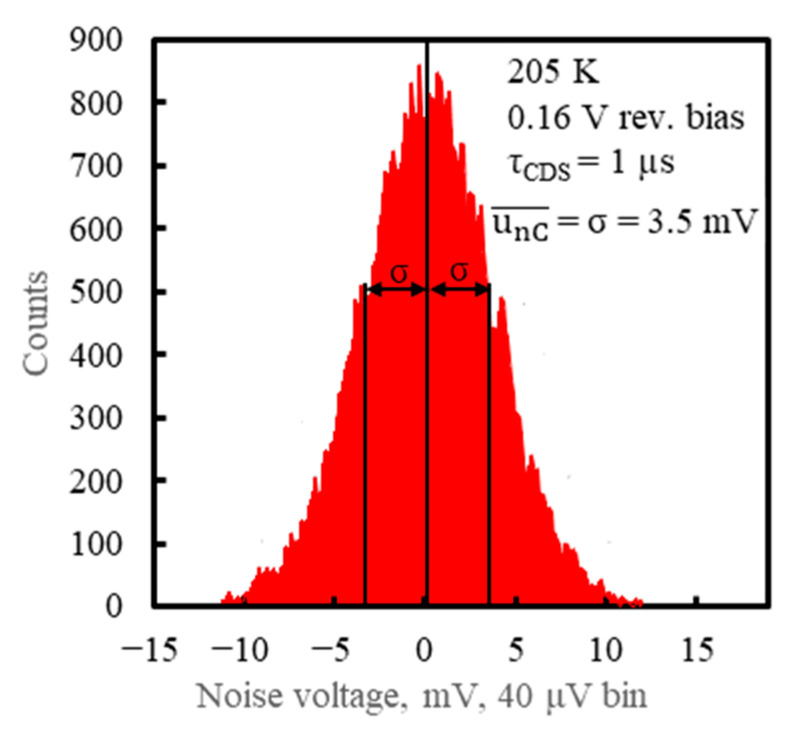
Histogram of the CDS noise voltage u_nC_, measured at the output C of the PSD module (adapted with permission from Ref. [20]; copyright: 2022, SPIE).

**Table 1 sensors-23-04915-t001:** Schematic diagram of the layers in the mesa structure: their doping concentration, material composition, and thickness.

	-	-	Indium	>6.0 µm	
CdTe—passivation	n^+^	N_D_ = 3 × 10^17^ cm^−3^	Hg_0.88_Cd_0.12_Te	1.5 µm	CdTe—passivation
	p^+^	N_A_ = 5 × 10^17^ cm^−3^	Hg_0.85_Cd_0.15_Te	0.7 µm	
	P^+^	N_A_ = 5 × 10^17^ cm^−3^	Hg_0.65_Cd_0.35_Te	0.8 µm	
	Graded bandgap	Doping gradient	Hg_1−x_Cd_x_Te	1.1 µm	
	p^−^–absorber	N_A_ = 2 × 10^15^ cm^−3^	Hg_0.80_Cd_0.20_Te	2.0 µm	
	Graded bandgap	Unintensional doping	Hg_1−x_Cd_x_Te	1.6 µm	
	N^+^	N_D_ = 3 × 10^17^ cm^−3^	Hg_0.68_Cd_0.32_Te	4.6 µm	
	Buffer	Unintensional doping	CdTe	2.7 µm	
	Substrate	semi-insulating	GaAs	400 µm	

**Table 2 sensors-23-04915-t002:** Differential resistance between the opposite cathode electrodes at ±16 mA, based on the results shown in Figure 9, at T_d_ = 205 K.

	R, Ω	ΔR = R − R_sh_, Ω	ΔR/R_sh_ = R/R_sh_ − 1
R_sh_	5.5	0.00	0.0%
R_C-B_ (−109 mV, −16 mA)	5.8	0.28	5.1%
R_C-B_ (+109 mV, +16 mA)	5.8	0.35	6.3%
R_D-A_ (−109 mV, −16 mA)	5.8	0.30	5.4%
R_D-A_ (+117 mV, +16 mA)	6.1	0.63	11.5%

**Table 3 sensors-23-04915-t003:** Output parameters of the PSD module, measured at the maximal difference between the signals from the opposite channels. The PSD module sample #1. No averaging. (adapted with permission from Ref. [20]; copyright: 2022, SPIE).

	Condition	Max (U_A_ − U_D_)	Max (U_D_ − U_A_)	Max (U_B_ − U_C_)	Max (U_C_ − U_B_)
Direct Measurand					
X, mm		−0.25	0.10	0.40	−0.55
Y, mm		0.40	−0.35	−0.10	−0.10
U_A_, V		0.97	0.46	0.37	0.36
$\bar{u_{nA}}$ rms, V, dark noise		3.6 × 10^−3^	3.6 × 10^−3^	3.6×10^−3^	3.6 × 10^−3^
U_B_, V		0.44	0.76	0.82	0.26
$\bar{u_{nB}}$ rms, V, dark noise		3.4 × 10^−3^	3.4 × 10^−3^	3.4 × 10^−3^	3.4 × 10^−3^
U_C_, V		0.67	0.56	0.30	0.87
$\bar{u_{nC}}$ rms, V, dark noise		3.5 × 10^−3^	3.5 × 10^−3^	3.5 × 10^−3^	3.5 × 10^−3^
U_D_, V		0.40	1.23	0.52	0.45
$\bar{u_{nD}}$ rms, V, dark noise		3.6 × 10^−3^	3.6 × 10^−3^	3.6 × 10^−3^	3.6 × 10^−3^
Indirect measurand					
U_A_ − U_D_, V		0.57	−0.78	−0.15	−0.09
(U_A_ − U_D_)/E_in_, V/J		2.2 × 10^9^	−3.0 × 10^9^	−5.7 × 10^8^	−3.5 × 10^8^
U_B_−U_C_, V		−0.23	0.20	0.52	−0.62
(U_B_ − U_C_)/E_in_, V/J		−8.8 × 10^8^	7.5 × 10^8^	2.0 × 10^9^	−2.4 × 10^9^
Y_PSD_ = (U_A_ − U_D_)/(U_A_ + U_D_) × 1, mm		0.41	−0.46	−0.17	−0.11
$\bar{y_{nPSD}}$ rms, µm		4.0	3.3	5.8	6.3
X_PSD_ = (U_B_ − U_C_)/(U_B_ + U_C_) × 1, mm		−0.21	0.15	0.47	−0.55
$\bar{x_{nPSD}}$ rms, µm		4.4	3.7	4.8	4.8

## Data Availability

Data available within the article.

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
