# Peer review of "LWIR Lateral Effect Position Sensitive HgCdTe Photodetector at 205 K"

_sensors, 2023, doi:10.3390/s23104915_

Round 1
Reviewer 1 Report
Attached Separately.

Reviewer 2 Report
In this work, Jaroslaw et al. have demonstrated the lateral effect position sensitive HgCdTe photodetector at 205 K. The device structure fabrication and the characterization of LWIR PSD has been discussed in detail. The results are interesting and useful but still there is a need of improvement in the manuscript. There are some comments mentioned below that are needed to be addressed.
1. In the introduction section, authors need to explain about LWIR HgCdTe PSD and should discuss about the previous reports.
2. The research gap should be clearer and the novelty should be discussed in the introduction section about LWIR HgCdTe PSD.
3. The motivation behind detecting 10-11 μm wavelength and 205 K operating temperature should be mentioned in the introduction.
4. How about the carrier concentrations of HgCdTe P+, HgCdTe absorber and HgCdTe N+?
5. Authors should show the light absorption spectra for the GaAs/HgCdTe P+/HgCdTe/HgCdTe N+ layer and compare with the spectral responsivity of the photodiode.
6. Figure 10: Why does the improvement in the 10-11 μm wavelength under varying reverse bias is not significant as compared to that of the 4-8 μm? Is it related to the material absorption properties?
7. Is there any correlation between the optimum applied reverse biases with the lateral position in the photodiode? Please explain.
Round 2
Reviewer 1 Report
The authors have incorporated most of the comments satisfactorily in their revised manuscript. Thus, it may be accepted for publication in its present form.